# Gastrin ameliorates heart failure and suppresses myocardial remodeling via the JAK2/STAT3 and ERK1/2 pathways

Yalu Du[1,2,3☯], Ning Wang[3☯], Jin Dong[3], Xiaohong Chai[3], Liang Liu[1,2], Haozhou Zhang[3], Bao Li[1,2*], Jinjing Yang[3*]

1 The Second Hospital of Shanxi Medical University,Taiyuan, Shanxi, China, 2 Shanxi Medical University, Taiyuan, Shanxi,China, 3 Department of Cardiology, Shanxi Key Laboratory of Heart Failure Precision Medicine, Shanxi Cardiovascular Hospital, Taiyuan, Shanxi, China

☯ These authors contributed equally to this work.
* 13623475425@163.com (BL); imyangjj@163.com (JY)

## Abstract

### Background

Previous studies have indicated elevated serum gastrin levels in individuals with HF. However, the association and underlying mechanisms between gastrin and HF remain unclear. This article aims to investigate the effects of gastrin on myocardial remodeling and HF, as well as its potential signal transduction mechanisms.

### Methods

In vivo studies were initially conducted to investigate the relationship between gastrin and HF, as well as the effects of gastrin on myocardial remodeling and HF. Gastrin levels were measured using ELISA kits to assess their association with ISO-induced HF. Echocardio- graphy, qRT-PCR analysis of hypertrophy (ANP, BNP, β-MHC) and fibrosis markers (COL1, COL3, α-SMA), hematoxylin-eosin staining, and Masson's trichrome staining were performed to evaluate the impact of gastrin on HF, MH, and fibrosis in mice.Furthermore, the effect of gastrin on cardiomyocyte hypertrophy was investigated in vitro using H9C2 cells, with F-actin staining and qRT-PCR analysis of ANP and BNP employed. Additionally, western blotting (WB) analysis of P-JAK2/ JAK2, P-STAT3/STAT3, and P-ERK/ERK in cardiac tissues and cells was used to identify pathways through which gastrin modulates HF and myocardial remodeling.

### Results

In vivo study, the ISO-treated mice exhibited significantly increased gastrin levels compared to the control group (P < 0.05). Furthermore, the ISO group showed significant cardiac hypertrophy, characterized by increased heart size, thickened ventricular walls, impaired cardiac function, and expanded fibrotic areas (P < 0.05). In contrast, the gastrin-only group exhibited no significant pathological changes.

**Data availability statement:** All relevant data are within the paper and its Supporting Information files.

**Funding:** This work was supported by the Fundamental Research Program of Shanxi Provice(No.202303021222362; 202403021211101) and Shanxi Cardiovascular Hospital Incentive Plan Fund(No. XYS20230102).

**Competing interests:** The authors have declared that no competing interests exist.

Co-treatment with gastrin and ISO notably attenuated these pathological changes, whereas CI-988(a CCK2R inhibitor) admini- stration partially reversed gastrin's protective effects (P < 0.05). In vitro study,the ISO group exhibited a significantly larger cardiomyocyte surface area and elevated expression of hypertrophy-associated biomarkers (ANP and BNP) compared to controls (P < 0.01). Gastrin treatment significantly suppressed these changes (P < 0.01), but this protective effect was partly reversed by the CCK2R antagonist CI-988 (P < 0.05). Additionally, phosphory-lation levels of JAK2, STAT3, and ERK were significantly increased in the ISO group (P < 0.05) both in mice cardiac tissues and H9C2 cells. Gastrin treatment sup-pressed these increases (P<0.05), an effect diminished by CI-988 (P < 0.05).

## Conclusions

Gastrin may exert protective effects against ISO-induced HF and myocardial remod-eling by inhibiting the JAK2/STAT3 and ERK1/2 pathways via the CCK2 receptor.

## Introduction

Heart failure (HF) refers to impaired cardiac function, where the pumping capacity of the heart is weakened. Consequently, it cannot adequately deliver nutrients to body systems, causing ischemia and hypoxia in multiple organs [1]. This condition results from various cardiovascular diseases, such as hypertension, diabetes, obesity, and hyperlipidemia, leading to significant hospital admissions [2]. Myocardial remodeling, characterized primarily by cardiac hypertrophy, fibrosis, and apoptosis, constitutes the essential pathological basis of HF [3]. Long-term cardiovascular stress prompts the secretion of various mediators, such as growth factors, cytokines, and vasoac-tive hormones, from cardiomyo- cytes and non-cardiomyocytes. These mediators subsequently promote hypertrophy, fibrosis, and inflammation. Cardiac fibroblasts mainly mediate fibrotic remodeling, with endothelial and immune cells also contribut-ing significantly [4]. Activation of the sympathetic nervous system critically promotes cardiac remodeling and HF progression. This activation leads to increased circulat-ing catecholamines and subsequent activation of β-adrenergic receptors (β-ARs) [5]. Previous studies have shown that chronic stimulation of myocardial β-ARs by isoproterenol (ISO), a β-AR agonist, can induce cytokines. These cytokines mediate myocardial injury and pathological remodeling [6]. Chronic β-AR activation has detri-mental cardiac effects in humans and experimental animal models, including cardiac fibrosis, hypertrophy, and impaired cardiac function [7,8]. Current treatments for HF include vasodilators to enhance oxygenation, diuretics to reduce cardiac workload, and modulators of β-adrenergic and angiotensin signaling pathways to counteract remodeling. Although these therapies have benefited some HF patients, not all patients respond effectively. Thus, there remains a need to develop new therapies that reverse cardiac fibrosis and hypertrophy [9]. Identifying regulatory molecules involved in HF and myocardial remodeling, as well as elucidating their mechanisms, holds considerable clinical significance.

Gastrin, a peptide hormone secreted by gastrin-secreting cells (G cells), predominantly located in the gastric antrum and duodenum, plays a critical role in stimulating gastric parietal cells (GPCs) to secrete hydrochloric acid (HCl) [10]. Its primary biological effects are mediated through binding to the cholecystokinin-2 receptor (CCK2R) on target cells [11]. Historically recognized as a key regulator of gastrointestinal epithelial development and tumorigenesis, gastrin has, in recent decades, been the subject of extensive research. Advances in biochemistry, immunochemistry, and molecular biology have significantly broadened the understanding of its functions beyond the traditional gastrointestinal context. Gastrin coordinates a complex network of signaling pathways involved in cellular proliferation and apoptosis [12]. Studies have shown chronic kidney disease patients exhibit elevated serum gastrin levels. Gastrin infusion alleviates renal injury and interstitial fibrosis in hypertensive nephropathy, independent of its role in regulating blood pressure [13]. Notably, both gastrin and its receptor, cholecystokinin receptor 2 (CCK2R), show abundant expression within cardiac tissues. Gastrin has demonstrated cardioprotective properties in rat models of myocardial ischemia/reperfusion injury [10] and is sug-gested to improve outcomes following myocardial infarction (MI) [14,15]. Empirical studies demonstrated that gastrin 17 or CCK2R agonist administration into coronary arteries in anesthetized pigs dose-dependently increases coronary blood flow and myocardial contractility [16]. Additionally, patients with sepsis-induced myocardial dysfunction (SMD) showed elevated plasma gastrin levels compared to those without SMD. Intravenous gastrin administration alleviated SMD and cardiac injury [17]. However, the involvement of gastrin in cardiac remolding and HF induced by ISO remains largely unexplored. This research aims to examine the influence of gastrin on ISO-triggered myocardial remodeling and the progression of HF, and further clarify the associated mechanisms. The results of this investigation could potentially identify novel therapeutic strategies and targets for managing HF.

Several intracellular signaling cascades are involved in the regulation of cardiac hypertrophy, including the calcium-dependent signaling pathway, cyclic guanosine monophosphate (cGMP)/protein kinase G (PKG) [18], PKC [19], mitogen-activated PK (MAPK) [20], and the Janus kinase/signal transducer and activator of transcription (JAK/STAT) pathway [21]. Among these, MAPKs are pivotal in mediating cardiac remodeling in response to pathological stress or increased physio-logical demand [22]. Specifically, extracellular signal-regulated kinase (ERK) 1/2 modulates cardiac growth by influencing cardiomyocyte elongation and expansion, thus determining whether hypertrophy develops eccentrically or concentrically [23, 24]. Additionally, JAK (primarily JAK2) and its downstream effector STAT (particularly STAT3) are upregulated during cardiac hypertrophy [25]. Fan [26] demonstrated that periplocymarin effectively inhibits MH by suppressing JAK2/STAT3 signaling, suggesting activation of this pathway promotes MH. ISO induces MH and significantly enhances ERK1/2 phosphorylation levels [27]. Chen [28] reported that pirfenidone exerts protective effects against ISO-induced myocar-dial inflammation by inhibiting JAK2/STAT3 signaling, thereby reducing hypertrophy. Thus, inhibiting the JAK2/STAT3 and ERK1/2 pathways effectively suppresses cardiomyocyte hypertrophy and myocardial remodeling. Signal molecules capable of such inhibition could represent promising therapeutic candidates for myocardial remodeling and HF treatment. It is of scientific interest to investigate whether gastrin modulates the progression of heart failure through regulation of the ERK1/2 and JAK2/STAT3 signaling pathways.

## Materials and methods

### 1. Animal experimentation

To minimize gender-related variability, sixty-five male C57BL/6 mice aged seven weeks and weighing 18–22 g were acquired from the National Institutes for Food and Drug Control in Beijing, China. Animals were maintained under stan-dardized conditions in the animal facility at Shanxi Institute of Medical and Life Sciences in Taiyuan, China. The mice were then randomly assigned to five experimental groups (n = 13 mice/group): control, gastrin treatment (30 μg/kg/day) [15], ISO administration (10 mg/kg/day) [29], ISO combined with gastrin, and ISO combined with gastrin plus CI-988 (300 μg/kg/day) [15]. Gastrin (MCE, China, purity: 99.83%) and CI-988 (Tocris Bioscience, UK, purity≥98%) were administered by intraperitoneal injection, while ISO (Sigma-Aldrich, USA, purity≥99.8%) was administered subcutaneously. One week after

an adaptation period, experimental groups received the respective treatments, while the control group received normal saline intraperitoneally. After two weeks of treatment, six mice per group were randomly selected and humanely euthanized by cervical dislocation, and their cardiac tissues were harvested for molecular analyses. The remaining seven mice per group continued treatments for an additional two weeks and subsequently underwent echocardiography and histopathological examinations. Dosages and durations of gastrin, ISO, and CI-988 treatments followed previously established protocols. This study received ethical approval from the Ethics Committee of Shanxi Provincial Cardiovascular Hospital(registration number: 2023kjt100; date:2023/12/1).

## 2. Echocardiography

Mice were positioned securely on a horizontal experimental platform under inhalational isoflurane anesthesia. To evaluate how gastrin affects ISO-triggered cardiac hypertrophy and heart failure, echocardiographic measurements were taken using a Vevo 3100 LT imaging system (Toronto, Canada) equipped with an MX400 ultrasound probe. An experienced operator acquired echocardiograms at the level of the left ventricle (LV) papillary muscles in both B-mode and M-mode. The collected parameters included fractional shortening (FS), ejection fraction (EF), cardiac output (CO), and LV posterior wall thickness at diastole (LVPWd). Data were averaged from three consecutive cardiac cycles for subsequent statistical analysis.

## 3. Animal samples handling

Following echocardiography, mice were weighed, and their body weights (BW) were recorded. Blood samples were collected via orbital sinus puncture before euthanasia by decapitation. Hearts were excised and rinsed thoroughly with normal saline to remove residual blood. Photographic documentation of the heart was performed. The heart weights (HW) and left tibial lengths (TL) were measured, after which HW/TL and HW/BW ratios were calculated. Myocardial tissue was collected as a transverse ring from the central LV area, immediately fixed in 4% paraformaldehyde, embedded in paraffin, and sectioned into 4-μm slices. These slices were utilized for histological examinations, including Masson's trichrome staining, hematoxylin-eosin (H&E) staining, immunohistochemistry, and immunofluorescence assays.All histological staining procedures were conducted in a blinded manner. Remaining myocardial tissue samples were snap-frozen and stored at −80°C for subsequent WB and quantitative real-time PCR (qRT-PCR) analyses.

## 4. H&E staining

Histological sections were subjected to deparaffinization and rehydration, followed by a one-minute incubation with high-definition staining pretreatment solution. Sequential hematoxylin and eosin staining were performed, and slides were subsequently dehydrated and coverslipped for microscopic visualization. Digitized images of stained sections were obtained using a pathology slide scanner (KF-PRO-005, KFBIO, Ningbo, China). Ten random fields per section were selected, and 40–60 cardiomyocytes per section were analyzed at 400×magnification to calculate the average cross-sectional area (CSA).

## 5. Masson's trichrome staining

Masson's trichrome staining was conducted to assess collagen accumulation within myocardial tissues after 28 days of treatment. Sections were dewaxed, hydrated, and stained following instructions provided by the Masson's Trichrome Staining Kit (G1006, Servicebio, China). Digitized scans of the stained tissue samples were generated using the KF-PRO-005 slide scanner (KFBIO, China). Morphometric analysis of each tissue section was performed to quantitatively assess myocardial fibrosis.

## 6. Immunohistochemical (IHC) staining

For IHC analysis of collagen I and III, tissue sections were first deparaffinized, rehydrated, and subjected to antigen retrieval via heating in Tris-EDTA (pH 8.0) buffer at 95°C for half an hour. Endogenous peroxidase activity was

inhibited with 3% hydrogen peroxide (H$_2$O$_2$) for 25 minutes, followed by blocking of nonspecific interactions using 3% goat serum at room temperature for 30 minutes. Sections were then incubated overnight at 4°C with rabbit monoclonal antibodies targeting collagen I (GB11022−3, 1:1000; Servicebio, China) and collagen III (GB111629−3, 1:500; Servicebio, China). Afterward, sections were incubated for 50 minutes at room temperature with secondary antibodies. Immunoreactivity was developed using 3′-diaminobenzidine (DAB) staining (G1212; Servicebio, China), and cell nuclei were counterstained with hematoxylin. Images of stained sections were obtained using a KF-PRO-005 slide scanning system (KFBIO, Ningbo, China). For quantitative assessment, ten microscopic fields from each section were randomly selected under 200 × magnification, analyzed with ImageJ software (version 1.42; NIH, Bethesda, USA), and collagen deposition was expressed as a percentage of collagen-positive area relative to the overall field.

## 7. Immunofluorescence (IF) staining

For α-SMA immunodetection, tissue sections were initially subjected to deparaffinization and subsequent rehydration. Next, antigen retrieval was facilitated by heating sections immersed in Tris-EDTA buffer solution (pH 8.0). Specifically, tissues underwent microwave irradiation at medium intensity for 10 min, followed by a resting interval of 5 min, another round of microwaving at a reduced-medium intensity for 5 min, a brief 2-minute pause, and an additional microwaving session at the same reduced-medium intensity for 5 min. After gradual cooling to room temperature, samples were washed thoroughly in phosphate-buffered saline (PBS) for 10 min. To minimize nonspecific immunoreactivity, slides were blocked using PBS supplemented with 3% bovine serum albumin (BSA) for 1 hour. Afterwards, sections were treated overnight at 4°C with anti-α-SMA rabbit monoclonal antibody (GB111364; 1:200 dilution; Servicebio, China). Following repeated PBS rinses, sections underwent incubation with secondary antibodies conjugated to fluorescein for 1 hour at ambient temperature. After another 10-minute rinse in PBS, nuclear staining was achieved by incubating sections with 4′,6-diamidino-2-phenylindole (DAPI) for 10 min, shielded from light. Confocal microscopy (FV31S-SW, Olympus, Japan) was used to acquire immunofluorescent images.

## 8. Cell culture experimental protocol

Rat cardiomyocyte line H9C2, provided by the Cell Bank of the Chinese Academy of Sciences, was routinely cultured in Dulbecco's Modified Eagle Medium (DMEM/F-12; GIBCO), enriched with 10% fetal bovine serum (Cellmax, China) and supplemented with antibiotics (1% penicillin-streptomycin). Cells were maintained in appropriate culture vessels under standard cell-culture conditions. Cardiomyocyte hypertrophy was induced through serum deprivation, using medium containing 1% fetal bovine serum for 18 hours [27], followed by stimulation with 10 µM ISO for 24 hours [30]. To determine gastrin's effect on ISO-stimulated hypertrophy, H9C2 cells were assigned to five groups: control, gastrin only (100 nM) [14], ISO alone, ISO combined with gastrin, and ISO combined with gastrin plus CI-988 (100 nM [14]. Gastrin, ISO, and CI-988 were administered simultaneously to the cells in relevant groups.

## 9. Cytoskeleton fluorescence staining

H9C2 cardiomyocytes were cultured on coverslips placed within 6-well plates. Upon establishment of the experimental conditions, cells were immobilized using 4% paraformaldehyde and permeabilized via exposure to 0.3% Triton X-100. Following PBS rinsing, cells were incubated with rhodamine-phalloidin solution (0.1%, CA1610; Solarbio, China) in the dark for half an hour. Subsequently, the coverslips containing stained cells were mounted onto microscope slides with an anti-fade medium containing DAPI. Morphological analysis was conducted by confocal laser scanning microscopy (FV31S-SW; Olympus, Japan), with representative images acquired at a magnification of 200 × . Fifty randomly selected cells from each group were measured using ImageJ software to determine cellular surface areas.

## 10. RNA extraction and qRT-PCR

Isolation of total RNA from H9C2 cells was performed with the SevenFast RNA Extraction Kit (SM132−02; Beijing, China), strictly adhering to the manufacturer's recommended procedures. For myocardial tissues, total RNA extraction was carried out using the Trizol-based approach. RNA purity and concentration were measured using the Nano-400A spectrophotometer (Allsheng, Hangzhou, China). Reverse transcription of RNA into complementary DNA (cDNA) was conducted in 20 μL reactions employing the PrimeScript RT Reagent Kit (TaKaRa, Japan), as per the provided instructions. The synthesized cDNA was diluted 5-fold before polymerase chain reaction (PCR) assays. Quantitative real-time PCR (qRT-PCR) analyses were executed on a Thermal Cycler 96 Real-Time PCR device (Thermo Fisher Scientific, Vantaa, Finland), with the total reaction volume set at 20 μL. PCR cycling involved initial denaturation at 95°C (30 sec), followed by 40 cycles comprising brief denaturation (95°C, 5 sec) and annealing-extension (60°C, 34 sec). Post-amplification, melting curve analyses were performed, after which the reactions were cooled to finalize the procedure. Primer sequences employed in PCR assays are detailed in Table 1 [31]. Expression data were normalized against the housekeeping gene 18S rRNA, and relative gene expression levels were calculated via the $2^{-\Delta\Delta CT}$ method.

## 11. WB

Cellular proteins from H9C2 and myocardial tissues from the left ventricle were isolated by lysing samples in RIPA buffer (AR0102–100; Boster Biotechnology, China) supplemented with protease and phosphatase inhibitor mixtures (AR117/1183; Boster Biotechnology, China). Left ventricular samples were mechanically disrupted with a Bio-Gen PRO200 homogenizer (USA) and further fragmented using an ultrasonic device (Ningbo, China). The homogenized samples were then centrifuged (14,000 × g, 4°C, 20 min) to separate soluble proteins, which were collected from the supernatants. Protein concentrations were determined through BCA assays. After adjusting to uniform protein amounts,

**Table1. Primers used in qRT-PCR.**

| Samples | genes | Primer sequence (5′ to 3′) |
|---------|-------|-----------------------------|
| Mouse | 18S forward | AAACGGCTACCACATCCAAG |
| | 18S reverse | TTGCCCTCCAATGGATCCT |
| | ANP forward | TCTTCCTCGTCTTGGCCTTT |
| | ANP reverse | CCAGGTGGTCTAGCAGGTTC |
| | BNP forward | TGGGAGGTCACTCCTATCCT |
| | BNP reverse | GGCCATTTCCTCCGACTTT |
| | Β-MHC forward | CGGACCTTGGAAGACCAGAT |
| | Β-MHC reverse | GACAGCTCCCCATTCTCTGT [31] |
| | CollagenI forward | CTGGCGGTTCAGGTCCAAT |
| | CollagenI reverse | TTCCAGGCAATCCACGAGC |
| | CollagenIII forward | CTGTAACATGGAAACTGGGGAAA |
| | CollagenIII reverse | CCATAGCTGAACTGAAAACCACC |
| | ACTA2 forward | GGACGTACAACTGGTATTGTGC |
| | ACTA2 reverse | TCGGCAGTAGTCACGAAGGA |
| Rat | 18S forward | CCGTTCTTAGTTGGTGGAGCGATT |
| | 18S reverse | TTGCTCAATCTCGGTGGCTGAAC |
| | ANP forward | GAAGATGCCGGTAGAAGATGAG |
| | ANP reverse | AGAGCCCTCAGTTTGCTTTTC |
| | BNP forward | GGTGCTGCCCCAGATGATT |
| | BNP reverse | CTGGAGACTGGCTAGGACTTC [31] |

samples were mixed with sample buffer, heated to 100°C for 8 min for denaturation, and separated using 8–10% SDS-PAGE. Subsequently, proteins were transferred onto PVDF membranes via semi-dry electrotransfer equipment (Bio-Rad). Membranes were then blocked at room temperature for 2 hours using either 5% skimmed milk or bovine serum albumin solution. Following blocking, membranes were probed overnight at 4°C with primary antibodies targeting JAK2 (CST; dilution 1:1000), phosphorylated JAK2 at residues Y1007 and Y1008 (Abcam; dilution 1:1000), STAT3 (CST; dilution 1:1000), phosphorylated STAT3 at Tyr705 (CST; dilution 1:2000), ERK (CST; dilution 1:1000), phosphorylated ERK1/2 at Thr202 and Tyr204 (CST; dilution 1:2000), and GAPDH (Abways; dilution 1:5000). After multiple washes with TBST buffer, membranes were incubated with appropriate secondary antibodies for 1 hour at ambient temperature. Protein bands were visualized using an ultra-sensitive ECL detection kit (Meilunbio, Dalian, China), with band intensities quantified using Image Lab software (Bio-Rad).

## 12. Statistical analysis

Data were first tested for normality and homogeneity of variance. Comparisons between two experimental conditions were carried out using Student's t-test. Whereas analysis involving more than two groups was performed by one-way analysis of variance (ANOVA) when the data met the assumptions of normality and homogeneity of variance, supplemented with Tukey's post hoc testing. When the assumption of homogeneity of variance was violated, Welch's one-way ANOVA was applied as an alternative. Data are presented as mean ± standard deviation (SD). Statistical significance was defined as a P-value below 0.05. All data processing and analyses were executed using GraphPad Prism (version 9.5).

## Results

### 1. ISO increases gastrin levels in mice, and gastrin alleviates ISO-induced HF via CCK2R

Blood samples from mice were collected by ocular enucleation at four weeks post-treatment to determine serum gastrin concentrations using an ELISA assay. Compared with control animals, ISO-treated mice exhibited notably elevated gastrin levels ($P < 0.05$, Fig 1A), implying that ISO may stimulate endogenous gastrin secretion. To investigate the potential role of elevated gastrin levels in ISO induced heart failure, consistent with prior experimental evidence [14,17,32],we administered exogenous gastrin supplementation.We found no evident differences were found between control and gastrin-only groups regarding cardiac dimensions. However, pronounced cardiac enlargement occurred in ISO-treated animals, whereas ISO-induced cardiac hypertrophy was significantly mitigated by gastrin co-treatment. Conversely, mice receiving ISO combined with gastrin plus CI-988 showed a marked reversal of gastrin's cardioprotective effects, demonstrating the involvement of CCK2R signaling (Fig 1B). Prior to euthanasia, cardiac function was evaluated via echocardiography (Fig 1C). Results indicated that CO, EF, and FS were similar between mice receiving gastrin alone and control animals. However, ISO significantly reduced these functional parameters ($P < 0.001$). Treatment with gastrin notably enhanced CO, EF, and FS in ISO-exposed animals ($P < 0.05$). Importantly, the beneficial influence of gastrin was markedly diminished when mice were concurrently treated with the selective CCK2R inhibitor CI-988 ($P < 0.05$; Fig 1D-F). Collectively, these observations suggest that gastrin exerts its protective effects against ISO-triggered heart failure via activation of the CCK2 receptor.

### 2. Gastrin mitigates ISO-induced MH; this effect is diminished by the CCK2R antagonist CI-988

Parameters including TL, ratios of heart weight relative to body weight (HW/BW), and heart weight relative to tibial length (HW/TL) were measured. Gastrin alone did not significantly alter these indices compared with controls. In contrast, ISO-treated animals showed significant elevations in HW/BW and HW/TL ratios ($P < 0.001$), whereas these increases were markedly alleviated by gastrin administration ($P < 0.01$). The aforementioned ratio showed a further increase in the group administered CI-988 in combination therapy($P < 0.05$; Fig 2A, B). Furthermore, ISO increased posterior left ventricular

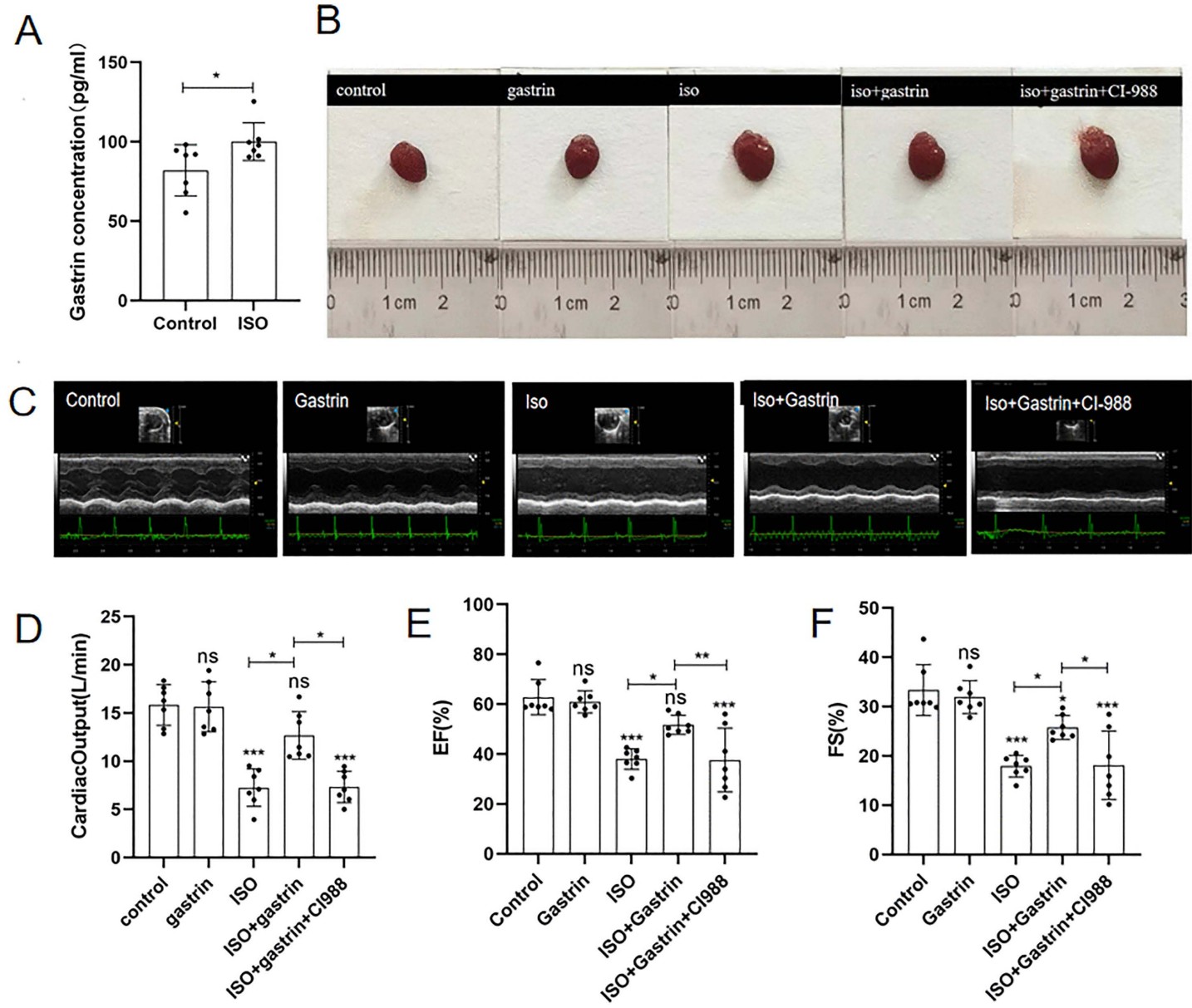

**Fig 1. Effects of gastrin on ISO-induced heart failure.** (A) Gastrin level of ISO and control group. (B) Comparison of heart size in each group of mice; (C)Echocardiography images of five groups; (D) Cardiac Output(CO) of five groups; (E) Left ventricular ejection fraction (EF) of five groups; (F) Left ventricular fraction shortening(FS) of five groups, n = 7.*P < 0.05, **P < 0.01,***P < 0.001, ****P < 0.0001. The data are presented as the mean±SEM. A student's t-test was performed for (Panel A), and one-way ANOVA with Tukey's post-hoc was conducted for (Panel B-F).

wall thickness (PLVW, P < 0.05), whereas gastrin reversed this effect significantly (P < 0.01). Similarly, the beneficial effects of gastrin on LVPW thickness were significantly reversed by CI-988 treatment (P < 0.01; Fig 2C). Analysis of cardiomyocyte cross-sectional areas (CSA), determined from histological evaluation of 50 cardiomyocytes per mouse following H&E staining (Fig 2E), indicated prominent cardiomyocyte enlargement upon ISO exposure (P < 0.001). Gastrin effectively reduced this hypertrophic response (P < 0.001), yet this reduction was partially negated when CI-988 was introduced simultaneously (P < 0.05; Fig 2D). Additionally, cardiac hypertrophy markers, including ANP, BNP, and β-MHC,

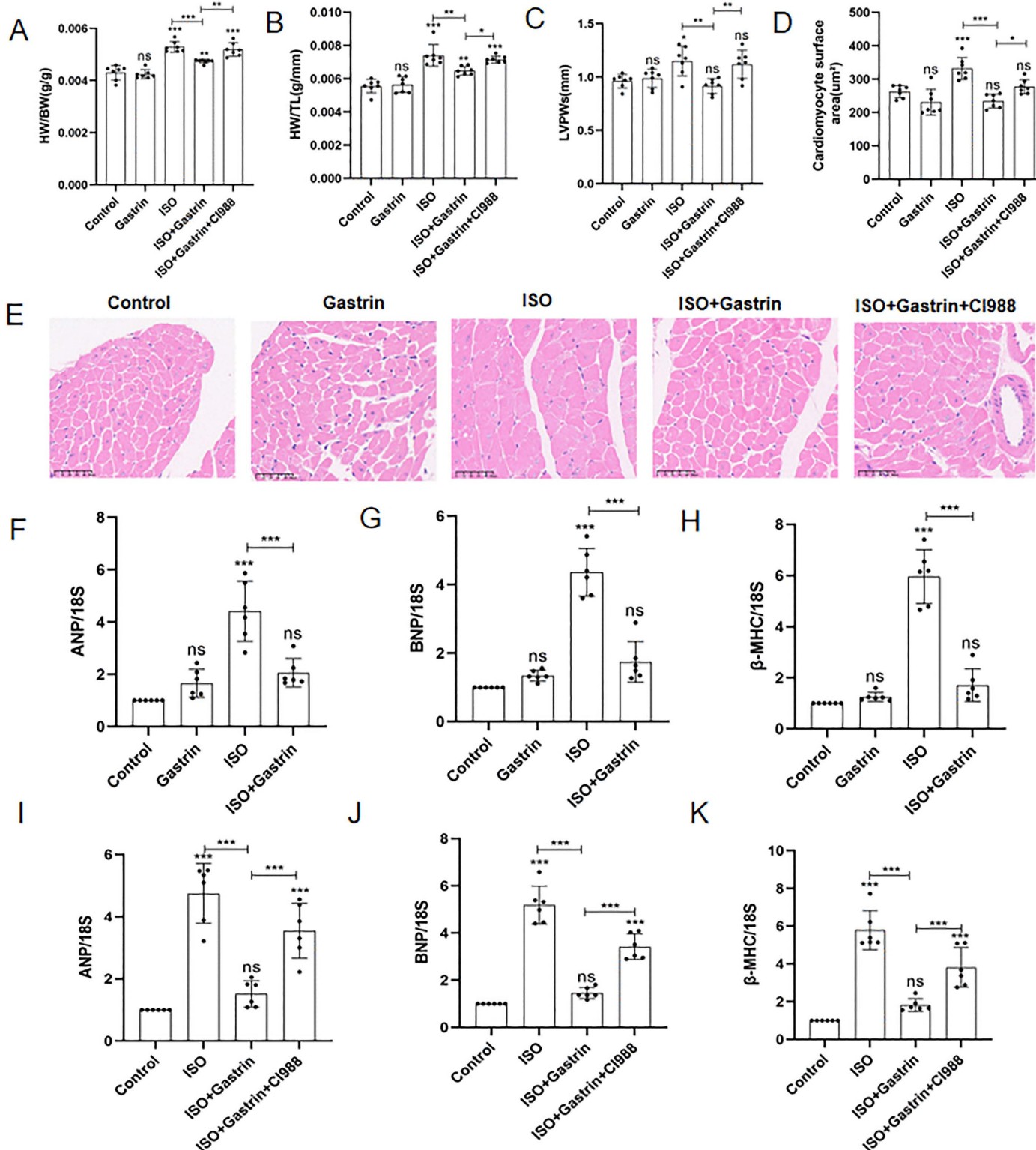

**Fig 2. Effects of gastrin on ISO-induced myocardial hypertrophy in vivo.** (A)Heart weight/Body weight(HW/BW) of five groups; (B)Heart weight/Tibial length(HW/TL) of five groups;(C)Posterior wall of left ventricle(LVPWs) of five groups; (D) Cardiomyocyte cross-sectional areas of five groups; n = 7. (E)Histological cross-sectional image of cardiomyocytes stained with hematoxylin and eosin (H&E);(F)(G)(H)Expression of ANP, BNP, and β-MHC in the

hearts of mice across Control, Gastrin, ISO, and ISO+Gastrin groups；(I)(J)(K)Expression of ANP, BNP, and β-MHC in the hearts of mice across Control, ISO, ISO+Gastrin and ISO+Gastrin+CI-988 groups.*P<0.05,**<0.01,***P<0.001, ****P<0.0001. The data are presented as the mean±SEM. One-way ANOVA with Tukey's post-hoc was conducted for (Panel A,C,D). Welch's one-way ANOVA analysis was performed of (Panel B,F-K).

were quantified using qRT-PCR [33]. ISO-treated animals showed significantly increased expression of these markers (P<0.001), whereas gastrin markedly suppressed their expression (P<0.001; Fig 2F-H). Co-administration with CI-988 reversed gastrin's inhibitory effect (P<0.001; Fig 2I-K). Collectively, these findings suggest that gastrin attenuates ISO-induced myocardial hypertrophy primarily through the activation of CCK2R, as the receptor antagonist CI-988 effectively diminished gastrin's protective impact.

### 3. Gastrin alleviates ISO-induced ventricular fibrosis via CCK2R

Masson's trichrome staining was applied to quantify myocardial fibrosis [34]. Compared with the control, fibrosis severity was markedly elevated in the ISO group (P<0.001), significantly attenuated by gastrin administration (ISO+gastrin group, P<0.001), but was again exacerbated in the ISO+gastrin+CI-988 group (P<0.05; Fig 3A, C). Immunohisto- chemical assays targeting collagen types I and III demonstrated similar tendencies, with substantial increases in collagen depo-sition in ISO-treated mice (P<0.001), a noticeable reduction upon gastrin intervention (P<0.01), and subsequent resto-ration of collagen accumulation after adding CI-988 (P<0.01; Fig 3B, D, E). Additionally, α-smooth muscle actin (α-SMA), a recognized marker of activated myofibroblasts [35], displayed comparable patterns: elevated expression in the ISO group, significantly decreased levels after gastrin treatment, and partial reversal upon CI-988 supplementation (Fig 4A). Consistent patterns were further confirmed by qRT-PCR analysis of collagen I, collagen III, and α-SMA mRNA expression in myocardial samples (P<0.05; Fig 4B-G). Collectively, these results highlight gastrin's protective role against ISO-induced myocardial fibrosis via activation of CCK2R signaling.

### 4. Gastrin alleviates HF and MH in mice by inhibiting the JAK2/STAT3 and ERK1/2 pathways

Accumulating evidence has demonstrated the involvement of JAK2/STAT3 and ERK1/2 pathways in MH [21,36–38]. In the current investigation, protein phosphorylation of JAK2, STAT3, and ERK significantly increased in the ISO group (P<0.01). No remarkable differences emerged between gastrin-only and control groups. Gastrin notably diminished the phosphor-ylation levels of JAK2, STAT3, and ERK induced by ISO (ISO+gastrin group, P<0.05; Fig 5A-D). Conversely, addition of the CCK2R antagonist CI-988 reversed gastrin's inhibitory effect (P<0.05; Fig 5E–H). These data suggest that gastrin's cardioprotective actions against MH and HF predominantly involve CCK2R-dependent suppression of JAK2/STAT3 and ERK1/2 signaling pathways.

### 5. Gastrin mitigates ISO-induced cardiomyocyte hypertrophy; this effect is attenuated by the CCK2R inhibitor CI-988

To investigate cardiomyocyte size alterations, H9C2 cytoskeletal structures were visualized by staining F-actin with Rhodamine-phalloidin. ISO treatment markedly increased cardiomyocyte size (P<0.001), while no significant changes occurred in the gastrin-alone group compared to controls. Notably, cardiomyocyte enlargement caused by ISO was reversed by gastrin administration (ISO+gastrin group, P<0.001). However, when CI-988 was co-administered, gastrin's inhibitory effect was significantly reversed, resulting in increased cell size (ISO+gastrin+CI-988 group, P<0.001; Fig 6A, B). Similarly, ANP and BNP mRNA expression in H9C2 cells increased dramatically with ISO treatment (P<0.01). Gastrin effectively reduced this ISO-induced upregulation (P<0.01; Fig 6C, D), whereas co-treatment with CI-988 weakened gas-trin's suppressive effect (P<0.05; Fig 6E, F). These results collectively confirm that gastrin attenuates ISO-induced H9C2 cardiomyocyte hypertrophy via mechanisms reliant upon CCK2R activation.

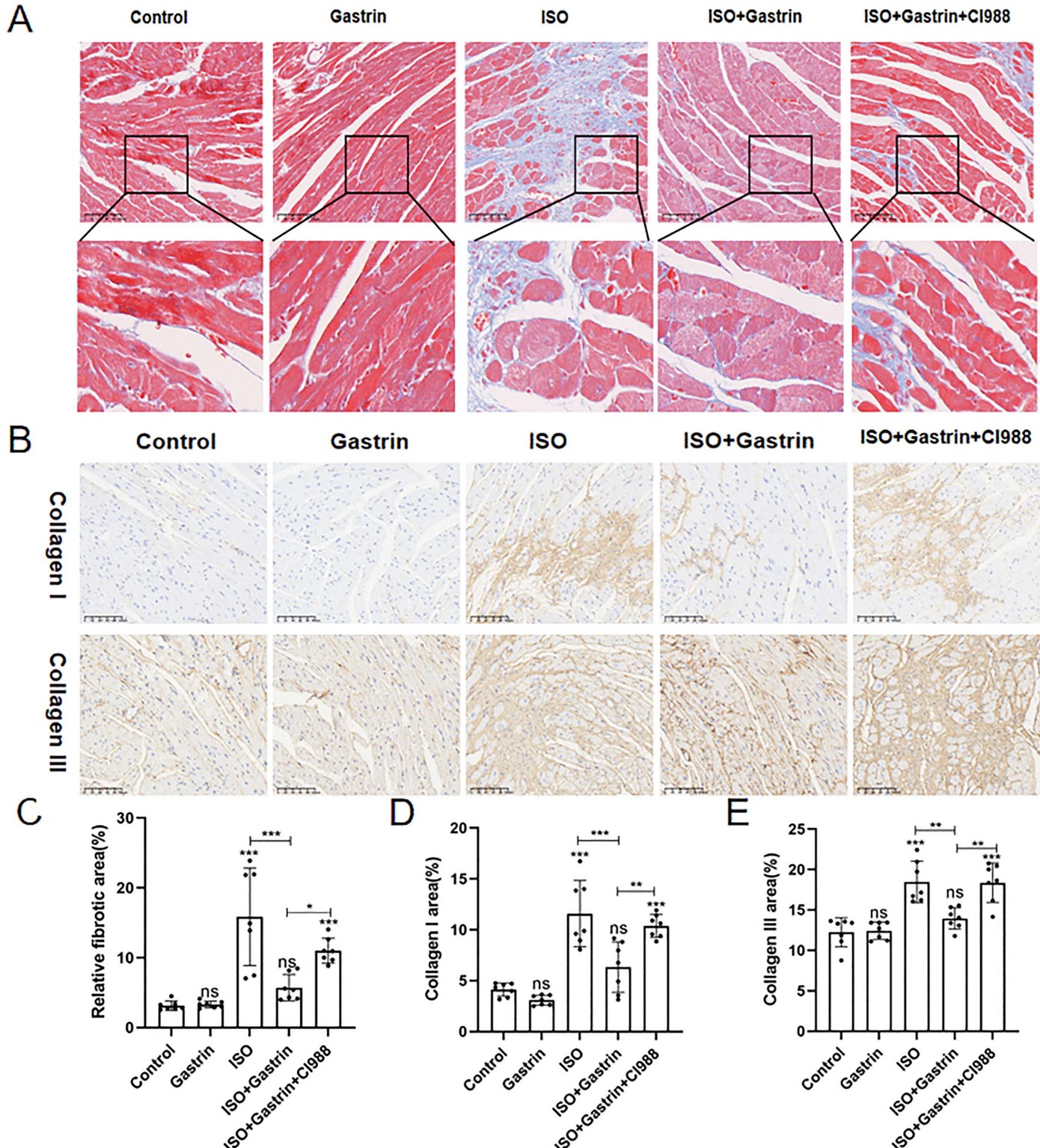

**Fig 3. The impact of gastrin on ISO-induced myocardial fibrosis in vivo(1).** (A)Masson trichrome staining images of each group(200x magnification); (B)Immunohistochemical staining images for collagen I and collagen III of each group(200x magnification); (C)The relative fibrotic area as demonstrated by Masson's trichrome staining; (D) Quantitative analysis of type I collagen areas; (E)Quantitative analysis of type III collagen areas.*P<0.05, **P<0.01, ***P<0.001, ****P<0.0001. The data are presented as the mean±SEM. One-way ANOVA with Tukey's post-hoc was conducted for (Panel C-E).

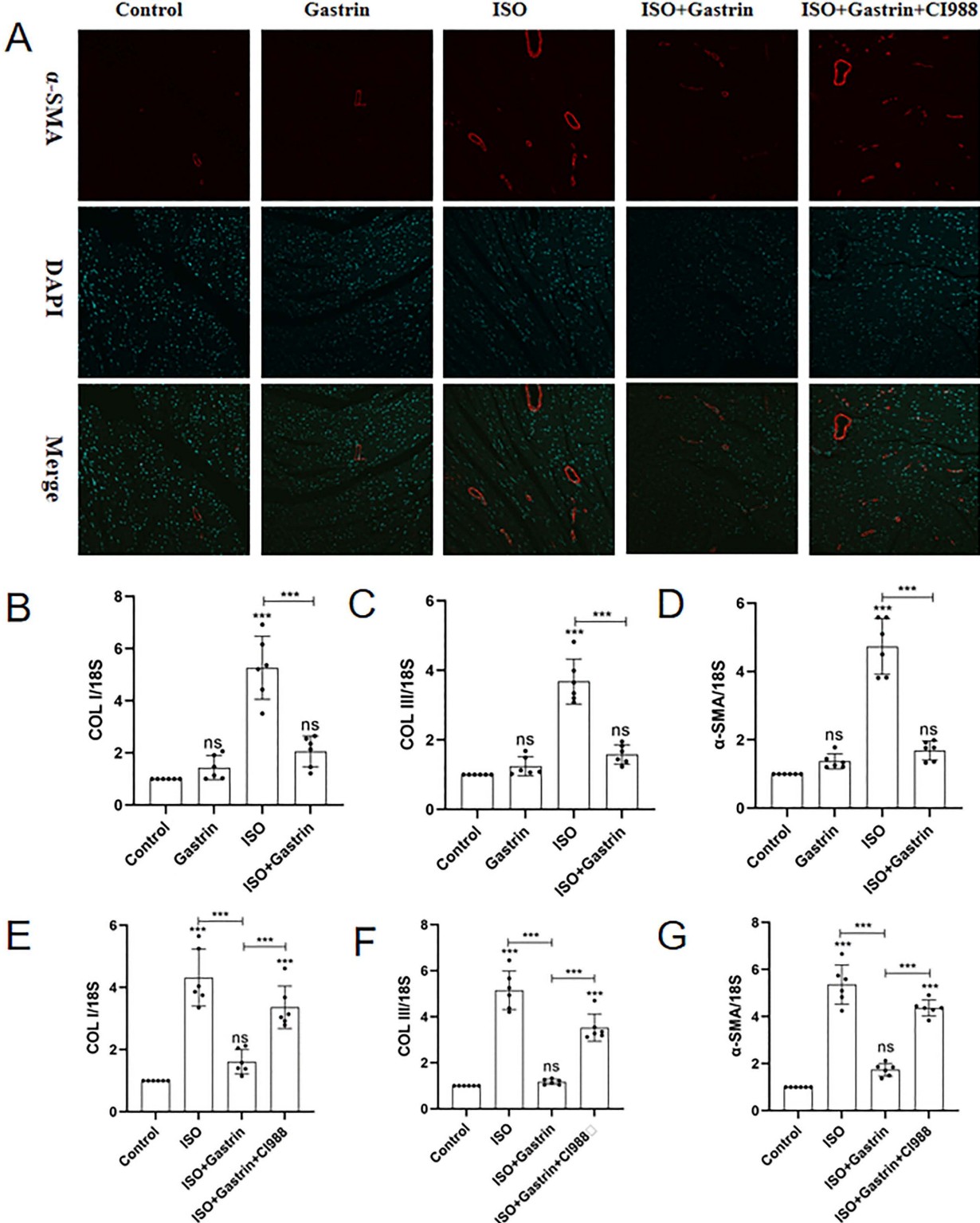

**Fig 4. The impact of gastrin on ISO-induced myocardial fibrosis in vivo(2).** (A)Immunofluorescence staining for alpha-smooth muscle actin (α-SMA); (B)(C)(D)The expression level of collagen I, collagen III and α-SMA in myocardial tissue of mice in the control, gastrin, ISO, and ISO+gastrin groups were quantified by qRT-PCR; (E)(F)(G)The expression level of collagen I, collagen III and α-SMA in myocardial tissue of mice in the control, ISO,

ISO+gastrin, and ISO+gastrin+CI-988 groups were quantified by qRT-PCR;*P<0.05, **P<0.01,***P<0.001, ****P<0.0001. The data are presented as the mean±SEM. Welch's one-way ANOVA analysis was performed of (Panel B-G).

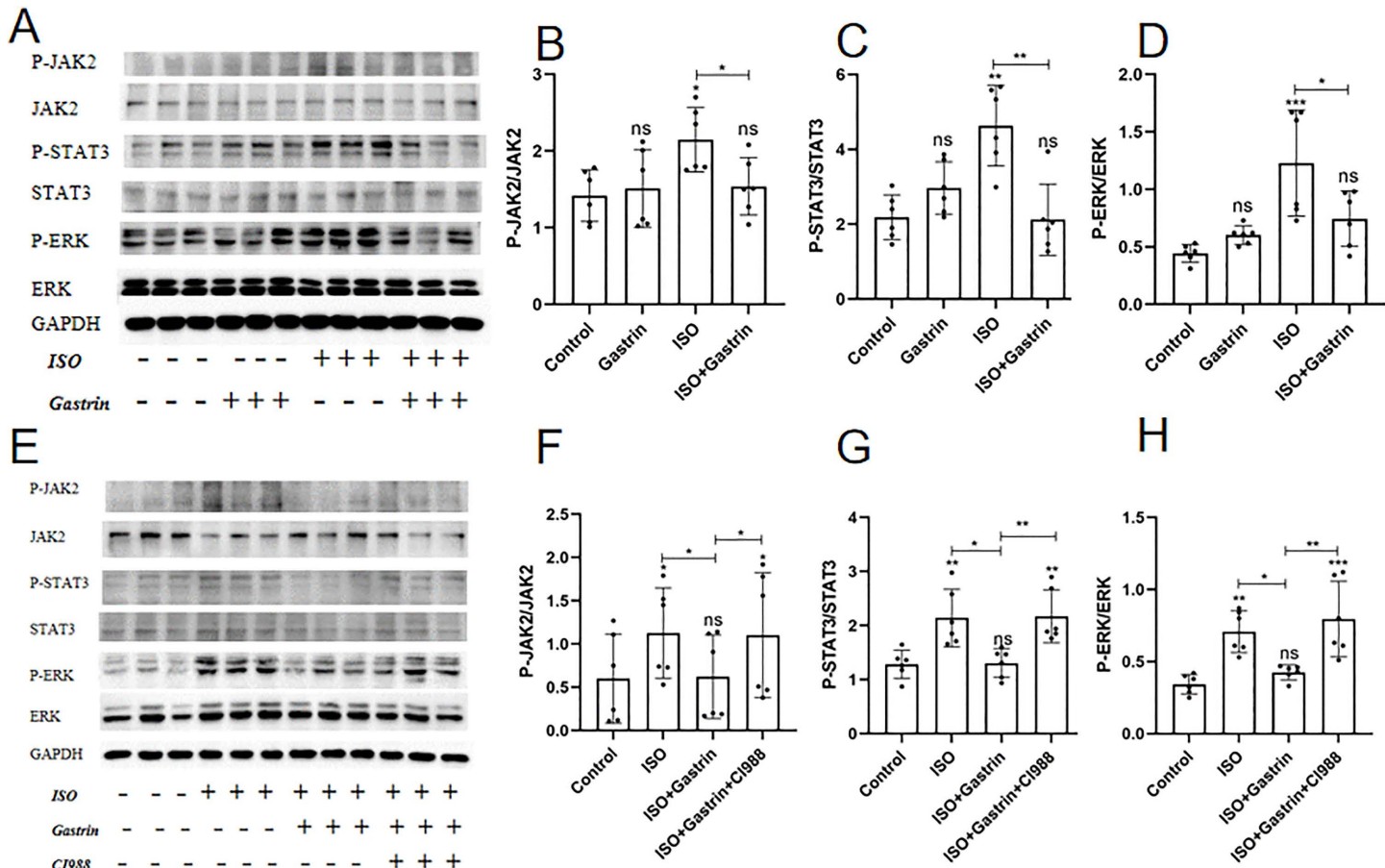

**Fig 5. The myocardial protective effect of gastrin through JAK2/STAT3 and ERK1/2 pathways in vivo.** In the control, gastrin, ISO, and ISO+-gastrin groups:(A)The protein expression levels of JAK2,p-JAK2, STAT3, p-STAT3, ERK1/2, (p-ERK, and GAPDH by Western blot; (B)The level of p-JAK2/JAK2; (C)The level of p-STAT3/STAT3; (D)The level of p-ERK/ERK; In the control, ISO, ISO+gastrin, and ISO+gastrin+CI-988 groups: (E)The protein expression levels of JAK2,p-JAK2, STAT3, p-STAT3, ERK1/2, p-ERK, and GAPDH by Western blot; (F)The level of p-JAK2/JAK2; (G)The level of p-STAT3/STAT3; (H)The level of p-ERK/ERK.*P<0.05, **P<0.01,***P<0.001, ****P<0.0001. The data are presented as the mean±SEM. One-way ANOVA with Tukey's post-hoc was conducted for (Panel B-D,F-H).

## 6. Gastrin attenuates ISO-induced cardiomyocyte hypertrophy by inhibiting the JAK2/STAT3 and ERK1/2 pathways

WB analysis was performed to evaluate phosphorylation of JAK2, STAT3, and ERK. Phosphorylation ratios (p-JAK2/JAK2, p-STAT3/STAT3, p-ERK/ERK) were significantly enhanced by ISO treatment (P<0.05), with no notable changes observed between gastrin-only and control groups. Co-treatment with gastrin effectively lowered these phosphorylation levels (ISO+gastrin group, P<0.05; Fig 7A-D), whereas CI-988 reversed gastrin's inhibitory effect, restoring phosphory-lation to higher levels (ISO+gastrin+CI-988 group, P<0.05; Fig 7E-H). Additionally, WB analysis confirmed that the ERK agonist mSIRK [39] and the STAT3 agonist Colivelin [40] activated p-ERK (Fig 7K, L) and p-STAT3 (P<0.05;Fig 7O, P).

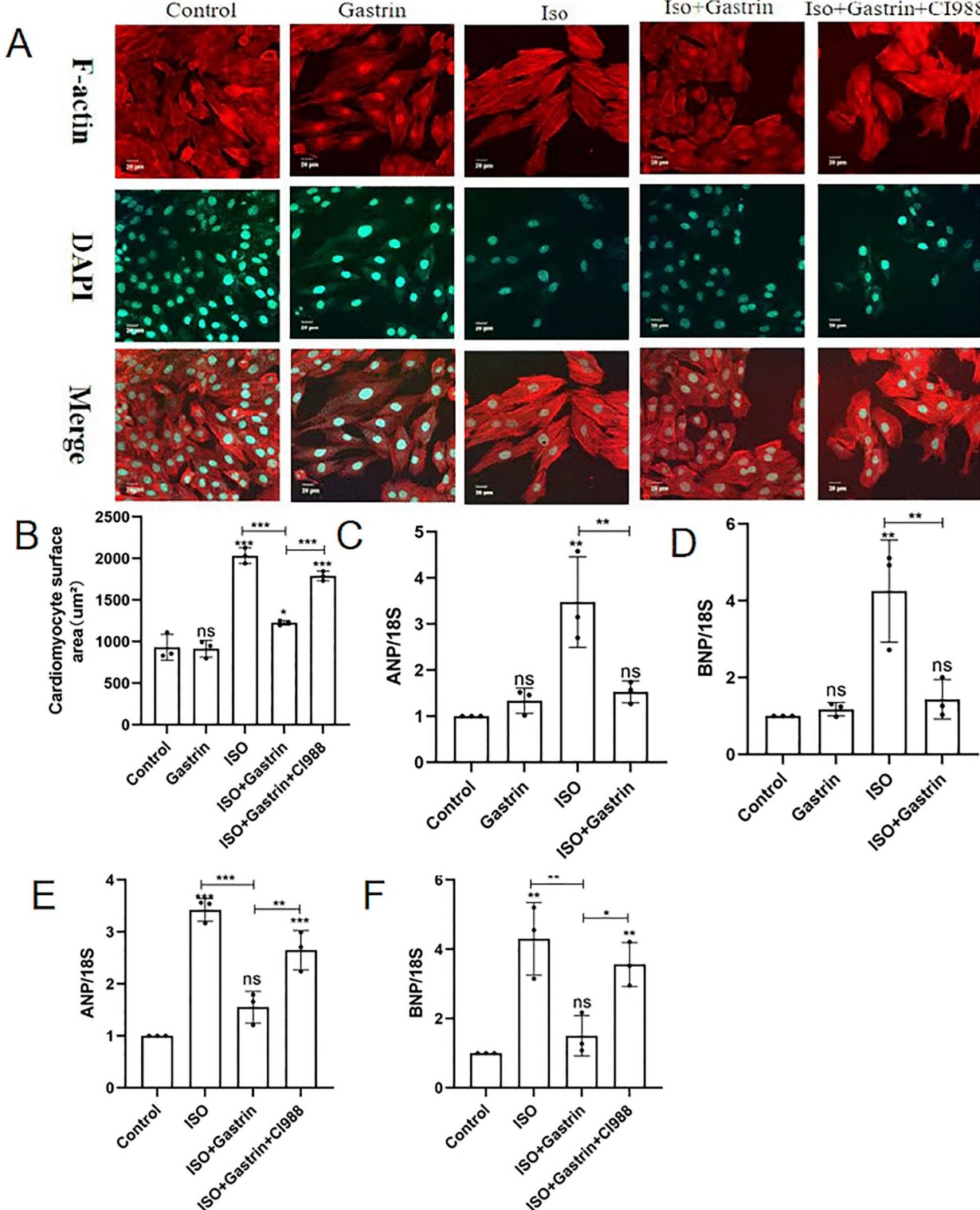

**Fig 6. Effect of gastrin on H9C2 cardiomyocyte hypertrophy.** (A)Rhodamine-phalloidin staining of F-actin(400×), DAPI stainded nuclei; (B) Quantificantion analysis of cardiomyocyte surface areas; (C-D) RT-PCR analysis for ANP and BNP mRNA in control, gastrin, ISO, and ISO+gastrin groups; (E-F) RT-PCR analysis for ANP and BNP mRNA in control, ISO, ISO+gastrin, and ISO+gastrin+ CI-988 groups;*P < 0.05, **P < 0.01,***P < 0.001,

****P<0.0001. The data are presented as the mean±SEM. One-way ANOVA with Tukey's post-hoc was conducted for (Panel B). Welch's one-wayANOVA analysis was performed of (Panle C-F).

Both agonists diminished gastrin's inhibitory effect on ISO-induced elevations in ANP and BNP (P<0.05; Fig 7I, J, M, N). Therefore, we propose that gastrin alleviates cardiomyocyte hypertrophy by suppressing the JAK2/STAT3 and ERK1/2 pathways.

## Discussion

We employed the traditional subcutaneous injection of isoproterenol to successfully induce the mouse model of myocardial hypertrophy and heart failure as previous study [29]. Our study represents the first evidence demonstrating a significant cardioprotective effect of gastrin in ISO-induced heart failure models. In vivo, gastrin administration markedly improved cardiac function and attenuated myocardial remodeling, including cardiomyocyte hypertrophy and interstitial fibrosis, in mice with ISO-induced heart failure. In vitro, gastrin effectively suppressed cardiomyocyte hypertrophy in ISO-stimulated cardiomyocytes. These protective effects are closely associated with gastrin's ability to inhibit excessive activation of the downstream JAK2/STAT3 and ERK1/2 signaling pathways through interaction with the CCK2R.

ISO induces a pathological state resembling heart failure by persistently activating β-adrenergic receptors, thereby mimicking chronic sympathetic overstimulation [41]. This condition is characterized by oxidative stress, inflammation, and robust activation of multiple pathological signaling pathways [42]. Previous studies have largely overlooked the role of gastrin in this experimental model. Our findings demonstrate that gastrin effectively attenuates ISO-induced cardiac injury, revealing a novel potential therapeutic approach for mitigating catecholamine-mediated cardiotoxicity.

Gastrin-like peptides have been identified in the myocardium across various animal species [43].Our study demonstrated increased gastrin levels in mice with HF, consistent with findings from numerous prior studies reporting increased gastrin concentrations in various cardiac dysfunction conditions. For example, Tania et al.[44] found elevated gastrin levels in HF patients. Fang [17] reported that plasma gastrin concentrations is increased in patients with sepsis-induced myocardial dysfunction (SMD).Similarly, Hypertensive nephropathy (HN) is associated with elevated serum gastrin [13]. Jiang [45] found hypertensive adults exhibited significantly increased serum gastrin compared to normotensive controls after consuming a mixed meal. Considering the role of increased gastrin play a protective role in the development of cardiac hypertrophy and cardiac dysfunction [17]. Thus, these findings implied that increased serum gastrin may serve as an early compensatory biomarker of cardiac dysfunction and heart failure.

Our results indicate that gastrin protects against HF, aligning with previous studies. Fu et al.[14]showed that gastrin enhances cardiac function after MI. Fang D et al.[17]demonstrated that intravenous gastrin administration alleviated SMD and cardiac injury. As well as, our findings indicate gastrin effectively reduces myocardial fibrosis, consistent with its inhibitory effects on renal fibrosis in hypertensive nephropathy [13] and cardiac fibrosis after MI. Moreover, our study reveals gastrin's ability to attenuate ISO-induced cardiomyocyte hypertrophy. Gastrin is known to promote the proliferation of cancer cells while exerting minimal effects on normal tissue cells [46,47].This selective biological response can be attributed to the conditional expression of the CCK2R, which is primarily induced during pathological or physiological processes such as inflammation or tissue repair [48]. While in this study, under ISO stimulation, gastrin was observed to inhibit myocardial hypertrophy in mice.

Our data demonstrate that ISO robustly activates the JAK2/STAT3 and the ERK1/2 signaling pathway, whereas gastrin treatment effectively counteracts this activation. The JAK2/STAT3 and the ERK1/2 pathway were well-established mediator of signal transduction in the development of myocardial hypertrophy and myocardial fibrosis [12].When cardiac sympathetic overload occurs, the cellular signaling pathways (JAK2/STAT3 and ERK1/2) is activated [49], leading to myocardial hypertrophy as a compensatory mechanism to counteract myocardial stress induced by increased cardiac load. However,

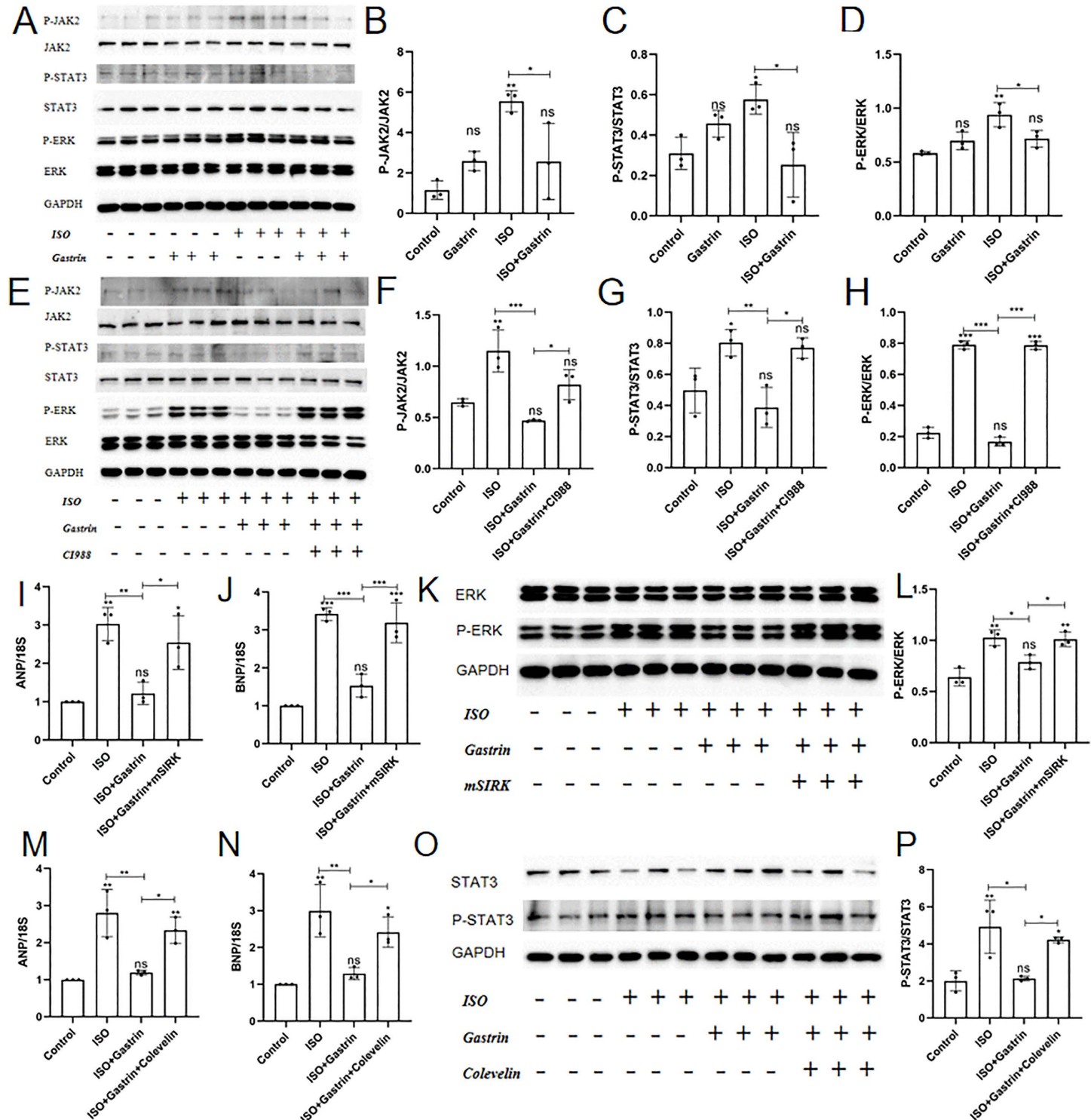

**Fig 7. Mechanism of gastrin in alleviating ISO-induced cardiomyocyte hypertrophy.** In the control,gastrin, ISO, and ISO+gastrin groups:(A)The protein expression levels of JAK2,p-JAK2, STAT3, p-STAT3, ERK1/2, p-ERK, and GAPDH by Western blot; (B)The level of p-JAK2/ JAK2; (C)The level of p-STAT3/ STAT3;(D)The level of p-ERK/ERK. In the control, ISO, ISO+ gastrin, and ISO+gastrin+ CI-988 groups: (E)The protein expression levels of JAK2,p-JAK2, STAT3, p-STAT3, ERK1/2, p-ERK, and GAPDH by Western blot;(F)The level of p-JAK2/ JAK2;(G)The level of p-STAT3/ STAT3;(H)The level of p-ERK/ERK. The expression of ANP(I),BNP(J), protein ERK1/2, p-ERK(K) and the ratio of p-ERK/ERK (L)after the addition of ERK agonist

mSIRK. The expression of ANP(M),BNP(N), protein STAT3,p-STAT3(O) and the ratio of p-STAT3/STAT3 (P)after the addition of STAT3 agonist colevelin.*P<0.05, **P<0.01. ***P<0.001. ****P<0.0001. The data are presented as the mean±SEM. One-way ANOVA with Tukey's post-hoc was conducted for (Panel B-D,F—H,L,P). Welch's one-wayANOVA analysis was performed of (Panel I,J,M,N).

sustained activation of this pathway may lead to eccentric myocardial hypertrophy, ultimately contributing to the development of heart failure [50]. Therefore, targeted inhibition of the hypertrophy signaling pathway may exert cardiopro- tective effects. Extensive evidence indicates gastrin mediates its biological effects via the CCK2R, engaging multiple downstream signaling cascades, including the MAPKs [51]and JAK2/STAT3 [52] signaling pathways.

In this study, we found that CI-988, a selective antagonist of CCR2R [53] treatment diminishes the inhibitory effect of gastrin in vivo and in vitro, this finding strongly suggests that gastrin exerts its inhibitory effect on myocardial remodeling, at least partially through activation of CCK2R and subsequent suppression of downstream JAK2/STAT3 and ERK1/2 signaling pathways. However, gastrin might mediate cardiac effects via the CCK2R, engaging multiple downstream signaling. For example, gastrin promoted angiogenesis and improved cardiac function in post-MI mice via regulating PI3K/Akt/vascular endothelial growth factor (VEGF) signaling pathway, and the protective effects were blocked by CI-988 [15]. Whereas PI3K/Akt/VEGF signaling is required for cardiac remodeling [54]. Thus, gastrin/CCK2R regulated PI3K/Akt/VEGF signaling might be responsible for ISO-induced cardiac remodeling, this warrants further experimental investigation.

The systemic toxic effects of ISO induce both cardiac hypertrophy and fibrosis. Gastrin demonstrates a distinct therapeutic advantage by coordinately inhibiting two key downstream pathogenic signaling pathways (JAK2/STAT3 and ERK1/2) through activation of a common upstream receptor, CCK2R, thereby achieving a "dual blockade" of the primary features of myocardial remodeling. This mechanism accounts for the comprehensive protective phenotype observed with gastrin treatment. Our findings indicate that targeting the gastrin-CCK2R axis may represent a promising therapeutic strategy for heart failure associated with sympathetic nervous system hyperactivity. The development of gastrin analogues or biased ligands for CCK2R, capable of selectively activating inhibitory signaling pathways while avoiding stimulatory effects, holds potential as a novel class of agents aimed at mitigating myocardial injury driven by neurohormonal dysregulation. Future studies should aim to validate these findings in additional models of sympathetic overactivation and further elucidate the underlying molecular mechanisms through which gastrin/CCK2R signaling modulates downstream pathways, including G protein-biased signaling, β-arrestin recruitment and Hippo-YAP signaling,etc.

This study has several limitations. First, the mechanism by which gastrin mediated cardiac remodeling was not thoroughly investigated. In the future, further experiments remained necessary. Second, we did not confirm whether CI-988-only affect cardiac hypertrophy in ISO-treated mice. Future research needs to investigate the role of CCK2R by using CI-988 or CCK2R knockout mice in ISO and/or TAC-induced cardiac remodeling model. Third, the anti-fibrotic effects of gastrin observed in vivo were not further dissected in an in vitro setting using cardiac fibroblasts. Future studies utilizing isolated fibroblast cultures are necessary to confirm the direct cellular target of gastrin. Finally, this study primarily focuses on the ISO model, and the generalizability of these findings to other heart failure models, such as those involving pressure overload or myocardial infarction, requires further investigation.

## Conclusion

Gastrin interacts with CCK2R to inhibit ISO-induced activation of the JAK2/STAT3 and ERK1/2 signaling pathways, thereby suppressing myocardial remodeling and improving cardiac function. These findings identify novel signaling mechanisms involving the gastrin/CCK2R/ JAK2/STAT3 and gastrin/CCK2R/ERK pathways,both pathways are critical for regulating cardiac remodeling and HF. These findings not only enhance our understanding of the physiological roles of gastrin but also potentially provide a foundation for developing new therapeutic strategies for cardiac remodeling-associated diseases.

## Supporting information

**S1 File. The original data and corresponding statistical values of the figures in the text.**
(DOCX)

**S2 File. Raw image.**
(DOCX)

**S3 File. Supplement supporitng information.**
(XLSX)

## Acknowledgments

We would like to extend our sincere gratitude to the Laboratory of Shanxi Provincial Cardiovascular Hospital for providing access to the experimental equipment and instruments essential for this study. Furthermore, we thank Professor Xiaoli Zhang from the Ultrasound Room of Shanxi Provincial Cardiovascular Hospital for her expert assistance during echocardiographic assessments in mice.

## Author contributions

**Conceptualization:** Bao Li, Jinjing Yang.

**Data curation:** Liang Liu.

**Formal analysis:** YaLu Du, Ning Wang.

**Funding acquisition:** YaLu Du, Jinjing Yang.

**Investigation:** YaLu Du, Ning Wang, Jin Dong.

**Methodology:** YaLu Du, Ning Wang, Bao Li, Jinjing Yang.

**Project administration:** Xiaohong Chai, Bao Li, Jinjing Yang.

**Resources:** Liang Liu.

**Software:** YaLu Du, Ning Wang, Haozhou Zhang.

**Supervision:** Jin Dong, Bao Li, Jinjing Yang.

**Validation:** Haozhou Zhang.

**Visualization:** Xiaohong Chai.

**Writing – original draft:** YaLu Du, Ning Wang.

**Writing – review & editing:** YaLu Du, Ning Wang, Jin Dong, Xiaohong Chai, Liang Liu, Haozhou Zhang, Bao Li, Jinjing Yang.

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
