## [Decision Letter · Decision Letter 0]

28 Sep 2025

Dear Dr. Li,

Thank you for submitting your manuscript to PLOS ONE. After careful consideration, we feel that it has merit but does not fully meet PLOS ONE’s publication criteria as it currently stands. Therefore, we invite you to submit a revised version of the manuscript that addresses the points raised during the review process.

We look forward to receiving your revised manuscript.

Kind regards,

Zhiling Yu

Academic Editor

PLOS ONE

7. Please remove your figures from within your manuscript file, leaving only the individual TIFF/EPS image files, uploaded separately. These will be automatically included in the reviewers’ PDF.

Additional Editor Comments:

1. Please specify the purity of the studied compound within the Materials and Methods section.

2. Please ensure the inclusion of the registration number and the full date (day, month, and year) of the ethical approval for conducting animal experiments.

3. Methods for animal anesthesia and euthanasia should be provided.

Reviewers' comments:

Reviewer's Responses to Questions

**Comments to the Author**

1. Is the manuscript technically sound, and do the data support the conclusions?

Reviewer #1: No

Reviewer #2: Yes

Reviewer #3: Partly

Reviewer #4: Yes

Reviewer #5: Yes

2. Has the statistical analysis been performed appropriately and rigorously?

Reviewer #1: No

Reviewer #2: I Don't Know

Reviewer #3: N/A

Reviewer #4: Yes

Reviewer #5: Yes

3. Have the authors made all data underlying the findings in their manuscript fully available?

Reviewer #1: No

Reviewer #2: Yes

Reviewer #3: Yes

Reviewer #4: Yes

Reviewer #5: Yes

4. Is the manuscript presented in an intelligible fashion and written in standard English?

Reviewer #1: No

Reviewer #2: Yes

Reviewer #3: Yes

Reviewer #4: Yes

Reviewer #5: Yes

Reviewer #1: REJECTED REJECTED REJECTED REJECTED REJECTED REJECTED VREJECTEDREJECTEDREJECTEDREJECTEDREJECTEDREJECTEDREJECTEDREJECTEDREJECTEDREJECTEDREJECTEDVREJECTEDREJECTEDREJECTEDREJECTEDREJECTEDREJECTEDREJECTEDREJECTEDREJECTED

Reviewer #2: This is an interesting study investigating the possible role of gastrin in a HF murine model. Overall, the study was in my opinion well-conducted but several issues made the reading of the manuscript often confusing.

The manuscript is not paginated and no line number are indicated making reference to parts of text difficult for the reviewer.

Abstract: The experimental design is not described making the understanding of results, difficult. What is CI-988? This is not explained.

Introduction: The gastrin part should be introduced earlier in the introduction. A long section talks about signalling in HF which is, in my opinion, unnecessary. The effects of various molecules on signalling pathway should not be described if it is not related to study. Maybe it is, but since the main player of this study, gastrin, has not been introduced, it is difficult for the reader to follow the intent of the authors.

It is mentioned that gastrin and its receptors are widely expressed in cell lines, animal and human samples without mentionning tissues or organs making this sentence useless.

The authors seems to imply that gastrin is secreted in pathological conditions. They also propose that gastrin has beneficial effects suggesting that this endogenous secretion is insufficient and an external imput is necessary.

Overall, the authors did not introduce gastrin in a way that links it to other roles than its traditional digestive roles.

Animal experimentation: I did not understand why half the animals were sacrificed after two weeks and another half after four weeks. I did not find an explanation for this elsewhere in the manuscript.

References should be included for dosages. A justification for studying only males should be included.

Results.

Again, I was not certain looking at the figure if I was looking at 2 or 4-week groups. Figure 1B seemed cropped on the left where control was written.

Starting in Figure 2, gene expression results were splitted into two. Ci-988 results being only in the Figures e to G. Only 3 samples/group are illustrated, which is very low. The authors should justify this since it does not correspond to what they described in the Methods. The same is in Figures, 4, 5 and 6. The last figure is on cell culture studies.

Discussion.

The authors should refrain to describe their model as a HF model since they did not look at other manifestations of the syndrome such as lung congestion or lower exercise capacity.

Many elements of the introduction should be in the discussion.

References: Correct #6, #9 and #32.

Overall, some work is needed to make this manuscript easier to read and to understand. The rationale is poorly explained leaving the reader sometimes lost.

Reviewer #3: Dr. Du studied the protective effect of gastrin supplementation in an ISO-induced heart failure model. They found that ISO treatment led to increased cardiac hypertrophy and impaired cardiac function due to enhanced fibrosis. Interestingly, ISO treatment also elevated serum gastrin levels compared to non-ISO-treated mice. The authors reported that additional gastrin supplementation attenuated cardiac hypertrophy and improved cardiac function in ISO-treated mice. Moreover, they observed that the protective effects of gastrin supplementation were partially reversed in the presence of CI-988, a CCK2R inhibitor. In contrast, the gastrin-only group exhibited no significant pathological changes. Immunoblotting further showed that activation of the JAK2, STAT3, and ERK pathways contributed to myocardial hypertrophy during ISO treatment. The authors concluded that gastrin supplementation provides benefits by inhibiting the JAK2/STAT3 and ERK1/2 pathways.

While this is an interesting study, the reviewer has some concerns.

1. The rationale for using additional gastrin in this study is unclear. ISO treatment alone already increased serum gastrin levels, indicating that ISO leads to elevated endogenous gastrin content. Administering a gastrin antagonist should be expected to worsen myocardial hypertrophy and cardiac function, and such results would more directly support the benefits of gastrin treatment in ISO-treated mice.

2. Animal groups. In this study, gastrin treatment alone was used as another control group. The reviewer is curious why the authors did not include a CI-988–only treatment group.

3. Statistical analysis

The authors stated that one-way ANOVA was used in this study. However, it is unclear whether all data passed tests for normality and equal variance. If not, what alternative statistical methods were applied?

4. Results

A major concern is the quality of the blots, particularly for phosphorylated proteins. In Figure 5A, the differences in p-JAK2 and p-STAT3 between groups are difficult to discern. Higher-quality images should be provided.

Figure 5E shows variability in GAPDH content between groups, particularly in the last five samples. It would be more appropriate to use total protein as a loading control.

In Figure 6, the authors need to specify the material used for this experiment.

Additionally, there are major quality differences between the Figure 6A and 6E panels for p-STAT3 and STAT3 images. The authors should explain these discrepancies.

5. Discussion

The authors state, “In this study, we established mouse models of ISO-induced MH and HF.” However, the ISO-induced heart failure model is already well established in the literature. What unique features does this model provide compared to previously reported studies?

“However, after consuming a mixed meal, hypertensive adults exhibited significantly increased serum gastrin compared to normotensive controls. Thus, gastrin may potentially serve as a biomarker for various chronic diseases.”

What do these statements mean?

Please discuss the selectivity of CI-988 and its potential off-target effects. Since CI-988 is a CCK2R antagonist, clarification is needed on whether it might interact with other receptors or signaling pathways that could influence the observed cardiac outcomes.

Reviewer #4: Dear Author,

The manuscript is very good. Congratulations on your work "Gastrin ameliorates heart failure and suppresses myocardial remodeling via the JAK2/STAT3 and ERK1/2 signaling pathways".

As I am not a native English speaker, I did not make any suggestions regarding spelling and agreement.

Reviewer #5: Gastrin is a peptide with gastrointestinal function. Du et al., report that gastrin has a cardioprotective effect and can be used as a potential therapeutic candidate. It activates CCK2 receptor and suppress JAK2/STAT3 and ERK1/2 signaling. This is a very strong manuscript with novel insights. The authors are suggested to address the following concerns:

Major Comments:

1. It is not clear whether the gastrin levels are comparable in human heart failure as observed in the ISO-treated mice.

2. The ISO model is short term. What would be the effect of gastrin on chronic pressure overload or post-myocardial infraction remodeling? This should be explored or at least addressed in discussion section.

3. Recent studies on fibroblast-specific signaling (Hippo-YAP) in remodeling can be discussed in context of gastrin.

4. STAT3 and ERK1/2 are known to be cardioprotective, whereas their inhibition appears to be protective in the manuscript. This could be context dependent regulation by gastrin, which should be discussed in detail.

5. While the authors have used CI-988, the manuscript does not provide any evidence on CCK2R localization or expression levels in fibroblasts or cardiomyocytes.

6. Discussion should address the limitation of only using cardiomyocytes and not cardiac fibroblasts for fibrosis experiments.

Minor Comments:

1. State in the methods, whether histological quantifications were blinded.

2. In abstract and discussion, the terms “hypertrophy reversal” and “functional recovery” should be used with clear distinction as these are not same.

3. The figures appear hazy; authors are suggested to upload good quality images according to the journal’s requirements.

**Do you want your identity to be public for this peer review?** For information about this choice, including consent withdrawal, please see our Privacy Policy

Reviewer #1: No

Reviewer #2: No

Reviewer #3: No

Reviewer #4: **Yes:** Natália Maria Maciel Guerra Silva

Reviewer #5: No

---

## [Author Response · Author response to Decision Letter 1]

8 Dec 2025

Question1:Please ensure that your manuscript meets PLOS ONE's style requirements, including those for file naming.

Response:The manuscript has been reformatted in accordance with the journal's submission guidelines.

Question2:Please confirm at this time whether or not your submission contains all raw data required to replicate the results of your study.

Response:All original data associated with the article have been compiled into the files titled "S1 file" and "Supplement to supporting Information" and have been submitted to the journal alongside this revised manuscript.

Question3:PLOS ONE now requires that authors provide the original uncropped and unadjusted images underlying all blot or gel results reported in a submission’s figures or Supporting Information files.

Response:All gel images presented in the manuscript have been saved in the file "S1_raw_images" in accordance with the journal's requirements.

Question4:When completing the data availability statement of the submission form, you indicated that you will make your data available on acceptance.

Response:I hereby acknowledge and accept the journal's data disclosure policy, which permits unrestricted and free access to all associated data.

Question5:PLOS requires an ORCID iD for the corresponding author in Editorial Manager on papers submitted after December 6th, 2016. Please ensure that you have an ORCID iD and that it is validated in Editorial Manager.

Response:The ORCID iD of the corresponding author BaoLi: 0009-0006 -4449-7743.

The ORCID record of the corresponding author BaoLi is https://orcid.org/ 0009-0006-4449-7743.

Question6:Please include your full ethics statement in the ‘Methods’ section of your manuscript file. In your statement, please include the full name of the IRB or ethics committee who approved or waived your study, as well as whether or not you obtained informed written or verbal consent. If consent was waived for your study, please include this information in your statement as well.

Response:The information regarding the ethics committee and the corresponding ethics approval number has been included in the "Methods" section. As follows:

This study received ethical approval from the Ethics Committee of Shanxi Provincial Cardiovascular Hospital(registration number: 2023kjt100; date:2023.12.1).(Line 179-181,Page 9)

Question7:Please remove your figures from within your manuscript file, leaving only the individual TIFF/EPS image files, uploaded separately. These will be automatically included in the reviewers’ PDF.

Response:The figures in the manuscript have been uploaded separately in accordance with the journal's requirements, and the figure captions have been incorporated into the manuscript.

Question8:If the reviewer comments include a recommendation to cite specific previously published works, please review and evaluate these publications to determine whether they are relevant and should be cited. There is no requirement to cite these works unless the editor has indicated otherwise.

Response:We gratefully acknowledge the editor for ensuring a fair and rigorous review process. We also appreciate the reviewers’ impartiality, as they did not request the inclusion of citations to their previously published works.

Response To Additional Editor Comments:

Question1:Please specify the purity of the studied compound within the Materials and Methods section.

Response:The purity of the compound under investigation has been documented in the manuscript.As follows:

Gastrin (MCE, China, purity: 99.83%) and CI-988 (Tocris Bioscience, UK, purity≥98%) were administered by intraperitoneal injection, while ISO (Sigma-Aldrich, USA, purity≥99.8%) was administered subcutaneously.(Line 166-169,Page 8)

Question2:Please ensure the inclusion of the registration number and the full date (day, month, and year) of the ethical approval for conducting animal experiments.

Response:The registration number and the full date (day, month, and year) of the ethical approval for conducting animal experiments have been contained in the manuscript.As follows:

This study received ethical approval from the Ethics Committee of Shanxi Provincial Cardiovascular Hospital(registration number: 2023kjt100; date:2023/12/1).(Line 178-181,Page 9)

Question3:Methods for animal anesthesia and euthanasia should be provided.

Response:Methods for animal anesthesia and euthanasia have been registered in the manuscript.As follows:

After two weeks of treatment, six mice per group were randomly selected and humanely euthanized by cervical dislocation, and their cardiac tissues were harvested for molecular analyses. (Line 172-174,Page 8)

---

## [Decision Letter · Decision Letter 1]

22 Dec 2025

Dear Dr. Li,

Thank you for submitting your manuscript to PLOS ONE. After careful consideration, we feel that it has merit but does not fully meet PLOS ONE’s publication criteria as it currently stands. Therefore, we invite you to submit a revised version of the manuscript that addresses the points raised during the review process.

We look forward to receiving your revised manuscript.

Kind regards,

Zhiling Yu

Academic Editor

PLOS One

Journal Requirements:

**Additional Editor Comments:**

Please adequately address the concerns raised by Reviewer 3.

Reviewers' comments:

Reviewer's Responses to Questions

**Comments to the Author**

Reviewer #2: All comments have been addressed

Reviewer #3: (No Response)

Reviewer #5: All comments have been addressed

2. Is the manuscript technically sound, and do the data support the conclusions?

Reviewer #2: Yes

Reviewer #3: No

Reviewer #5: Yes

3. Has the statistical analysis been performed appropriately and rigorously?

Reviewer #2: Yes

Reviewer #3: (No Response)

Reviewer #5: Yes

4. Have the authors made all data underlying the findings in their manuscript fully available?

Reviewer #2: Yes

Reviewer #3: (No Response)

Reviewer #5: Yes

5. Is the manuscript presented in an intelligible fashion and written in standard English?

Reviewer #2: Yes

Reviewer #3: (No Response)

Reviewer #5: Yes

Reviewer #2: My comments were addressed appropriately. The manuscript is easier to read and figures are now more convincing.

Reviewer #3: I double checked the author's response and found that my previous concerns have not been properly addressed.

Reviewer #5: (No Response)

**Do you want your identity to be public for this peer review?** For information about this choice, including consent withdrawal, please see our Privacy Policy

Reviewer #2: **Yes:** Jacques Couet

Reviewer #3: No

Reviewer #5: No

---

## [Author Response · Author response to Decision Letter 2]

27 Jan 2026

We have carefully addressed all comments raised by Reviewer 3 and incorporated the corresponding revisions into the manuscript.

---

## [Decision Letter · Decision Letter 2]

5 Feb 2026

Gastrin ameliorates heart failure and suppresses myocardial remodeling via the JAK2/STAT3 and ERK1/2 signaling pathways

PONE-D-25-30912R2

Dear Author,

We’re pleased to inform you that your manuscript has been judged scientifically suitable for publication and will be formally accepted for publication once it meets all outstanding technical requirements.

Kind regards,

Zhiling Yu

Academic Editor

PLOS One

Additional Editor Comments (optional):

Reviewers' comments:

Reviewer's Responses to Questions

**Comments to the Author**

Reviewer #2: All comments have been addressed

Reviewer #3: All comments have been addressed

Reviewer #5: All comments have been addressed

2. Is the manuscript technically sound, and do the data support the conclusions?

Reviewer #2: Yes

Reviewer #3: Yes

Reviewer #5: Yes

3. Has the statistical analysis been performed appropriately and rigorously?

Reviewer #2: Yes

Reviewer #3: Yes

Reviewer #5: I Don't Know

4. Have the authors made all data underlying the findings in their manuscript fully available?

Reviewer #2: Yes

Reviewer #3: Yes

Reviewer #5: Yes

5. Is the manuscript presented in an intelligible fashion and written in standard English?

Reviewer #2: Yes

Reviewer #3: Yes

Reviewer #5: Yes

Reviewer #2: The authors had already addressed my comments in their first revision. The additions made in the current version seem appropriate.

Reviewer #3: This is the much improved version. All of my previous concerns have been addressed. There are no further comments from the reviewer.

Reviewer #5: (No Response)

**Do you want your identity to be public for this peer review?** For information about this choice, including consent withdrawal, please see our Privacy Policy

Reviewer #2: **Yes:** Jacques Couet

Reviewer #3: No

Reviewer #5: No

---

## [Editor Report · Acceptance letter]

PONE-D-25-30912R2

PLOS One

Dear Dr. Li,

I'm pleased to inform you that your manuscript has been deemed suitable for publication in PLOS One. Congratulations! Your manuscript is now being handed over to our production team.

Kind regards,

on behalf of

Dr. Zhiling Yu

Academic Editor

PLOS One